# GRAPH-AUGMENTED NORMALIZING FLOWS FOR ANOMALY DETECTION OF MULTIPLE TIME SERIES

**Enyan Dai**[*]
Pennsylvania State University
emd5759@psu.edu

**Jie Chen**[†]
MIT-IBM Watson AI Lab, IBM Research
chenjie@us.ibm.com

## ABSTRACT

Anomaly detection is a widely studied task for a broad variety of data types; among them, multiple time series appear frequently in applications, including for example, power grids and traffic networks. Detecting anomalies for multiple time series, however, is a challenging subject, owing to the intricate interdependencies among the constituent series. We hypothesize that anomalies occur in low density regions of a distribution and explore the use of normalizing flows for unsupervised anomaly detection, because of their superior quality in density estimation. Moreover, we propose a novel flow model by imposing a Bayesian network among constituent series. A Bayesian network is a directed acyclic graph (DAG) that models causal relationships; it factorizes the joint probability of the series into the product of easy-to-evaluate conditional probabilities. We call such a graph-augmented normalizing flow approach GANF and propose joint estimation of the DAG with flow parameters. We conduct extensive experiments on real-world datasets and demonstrate the effectiveness of GANF for density estimation, anomaly detection, and identification of time series distribution drift.

## 1 INTRODUCTION

Anomaly detection (Pimentel et al., 2014; Ruff et al., 2018) is the task of identifying unusual samples that significantly deviate from the majority of the data instances. It is applied in a broad variety of domains, including risk management (Aven, 2016), video surveillance (Kiran et al., 2018), adversarial example detection (Grosse et al., 2017), and fraud detection (Roy & George, 2017). Representative classical methods for anomaly detection are one-class support vector machines (Schölkopf et al., 2001) and kernel density estimation (Parzen, 1962; Kim & Scott, 2012). These methods rely on handcrafted features and often are not robust for high-dimensional data (e.g., images, speech signals, and time series). In recent years, inspired by the success of deep learning for complex data, many deep anomaly detection methods were proposed and they are remarkably effective in applications (Ruff et al., 2018; Sabokrou et al., 2018; Goyal et al., 2020).

Apart from these demonstrated applications, increasing demand exists for the anomaly detection of even more complex data; i.e., multiple time series. They contain a set of multivariate time series that often interact with each other in a system. A prominent example of the source of multiple time series is the power grid, where each constituent series is the grid state over time, recorded by a sensor deployed at a certain geographic location. The grid state includes many attributes; e.g., current magnitude and angle, voltage magnitude and angle, and frequency. Time series readings from sensors at nearby locations are often correlated and their behavior may be causal under cascading effects. Anomaly detection amounts to timely identifying abnormal grid conditions such as generator trip and insulator damage.

Anomaly detection of multiple time series is rather challenging, due to high dimensionality, interdependency, and label scarcity. First, a straightforward approach is to concatenate the constituent series along the attribute dimension and apply a detection method for multivariate time series. However, when the system contains many constituents, the resulting data suffer high dimensionality. Second, constituent series bear intricate interdependencies, which may be implicit and challenging to model. When an explicit graph topology is known, graph neural networks are widely used to digest the rela-

---

[*]This work was done while E. Dai was an intern at MIT-IBM Watson AI Lab, IBM Research.
[†]To whom correspondence should be addressed.

tional information (Seo et al., 2016; Li et al., 2018b; Yu et al., 2018; Zhao et al., 2019). However, a graph may not always be known (because, for example, it is sensitive information) and hence graph structure learning becomes an indispensable component of the solution (Kipf et al., 2018; Wu et al., 2020; Shang et al., 2021; Deng & Hooi, 2021). Third, labeling information is often limited. Even if certain labels are present, in practice, many anomalies may still stay unidentified because labeling is laborious and expensive. Hence, unsupervised approaches are the most suitable choice. However, albeit many unsupervised detection methods were proposed (Ruff et al., 2018; Sabokrou et al., 2018; Malhotra et al., 2016; Hendrycks et al., 2019), they are not effective for multiple time series.

In this work, we explore the use of normalizing flows (Dinh et al., 2016; Papamakarios et al., 2017) for anomaly detection, based on a hypothesis that anomalies often lie on low density regions of the data distribution. Normalizing flows are a class of deep generative models for learning the underlying distribution of data samples. They are unsupervised and they resolve the label scarcity challenge aforementioned. An advantage of normalizing flows is that they are particularly effective in estimating the density of any sample. A recent work by Rasul et al. (2021) extends normalizing flows for time series data by expressing the density of a series through successive conditioning on historical data and applying conditional flows to learn each conditional density, paving ways to build sophisticated flows for multiple time series.

We address the high dimensionality and interdependency challenges by learning the relational structure among constituent series. To this end, Bayesian networks (Pearl, 1985; 2000) that model causal relationships of variables are a principled choice. A Bayesian network is a directed acyclic graph (DAG) where a node is conditionally independent of its non-descendents given its parents. Such a structure allows factorizing the intractable joint density of all graph nodes into a product of easy-to-evaluate conditional densities of each node. Hence, learning the relational structure among constituent series amounts to identifying a DAG that maximizes the densities of observed data.

We propose a novel framework, GANF (Graph-Augmented Normalizing Flow), to augment a normalizing flow with graph structure learning and to apply it for anomaly detection. There are non-trivial technical problems to resolve to materialize this framework: (i) How does one inject a graph into a normalizing flow, which essentially maps one distribution to another? (ii) How does one learn a DAG, which is a discrete object, inside a continuous flow model? The solution we take is to factorize the density of a multiple time series along the attribute, the temporal, and the series dimensions and use a graph-based dependency encoder to model the conditional densities resulting from factorization. Therein, the graph adjacency matrix is a continuous variable and we impose a differentiable constraint to ensure that the corresponding graph is acyclic (Zheng et al., 2018; Yu et al., 2019). We propose a joint training algorithm to optimize both the graph adjacency matrix and the flow parameters.

In addition to resolving the high dimensionality and interdependency challenges, an advantage of modeling the relational graph structure among constituent series is that one can easily observe the dynamics of the data distribution from the graph. For time series datasets that span a long period, one naturally questions if the distribution changes over time. The graph structure is a useful indicator of distribution drift. We will study the graph evolution empirically observed.

We highlight the following contributions of this work:

- We propose a framework to augment a normalizing flow with graph structure learning, to model interdependencies exhibited inside multiple time series.

- We apply the augmented flow model to detect anomalies in multiple time series data and perform extensive empirical evaluation to demonstrate its effectiveness on real-world data sets.

- We study the evolution of the learned graph structure and identify distribution drift in time series data that span a long time period.

## 2 RELATED WORK

**Anomaly Detection.** Anomaly detection is a widely studied subject owing to its diverse applications. Recently, inspired by the success of deep learning, several deep anomaly detection methods are proposed and they achieve remarkable success on complex data, such as individual time series (Malhotra et al., 2016), images (Sabokrou et al., 2018), and videos (Ionescu et al., 2019). These

methods generally fall under three categories: deep one-class models, generative model-based methods, and transformation-based methods. Deep one-class models (Ruff et al., 2018; Wu et al., 2019) treat normal instances as the target class and identify instances that do not belong to this class. In generative model-based methods (Malhotra et al., 2016; Nguyen et al., 2019; Li et al., 2018a), an autoencoder or a generative adversarial network is used to model the data distribution. Then, an anomaly measure is defined, such as the reconstruction error in autoencoding. Transformation-based methods (Golan & El-Yaniv, 2018; Hendrycks et al., 2019) are based on the premise that transformations applied to normal instances can be identified while anomalies not. Various transformations such as rotations and affine transforms have been investigated. On the other hand, anomaly detection of multiple time series is under explored. Recently, Deng & Hooi (2021) study the use of graph neural networks in combination with structure learning to detect anomalies. Our method substantially differs from this work in that the learned structure is a Bayesian network, which allows density estimation. Moreover, the Bayesian network identifies conditional dependencies among the constituent series and induces a better interpretation of the graph as well as the data distribution.

**Normalizing Flows.** Normalizing flows are generative models that normalize complex real-world data distributions to "standard" distributions by using a sequence of invertible and differentiable transformations. Dinh et al. (2016) introduce a widely used normalizing flow architecture—RealNVP—for density estimation. Various extensions and improvements are proposed (Papamakarios et al., 2017; Hoogeboom et al., 2019; Kingma & Dhariwal, 2018). For example, Papamakarios et al. (2017) view an autoregressive model as a normalizing flow. To model temporal data, Rasul et al. (2021) use sequential models to parameterize conditional flows. Moreover, graph normalizing flows are proposed to handle graph structured data and improve predictions and generations (Liu et al., 2019). In contrast, normalizing flows for multiple time series are rarely studied in the literature. In this work, we develop a graph-augmented flow for density estimation and anomaly detection of multiple time series.

## 3 PRELIMINARIES

We first recall key concepts and familiarize the reader with notations to be used throughout the paper.

### 3.1 NORMALIZING FLOWS

Let $\mathbf{x} \in \mathbb{R}^D$ be a $D$-dimensional random variable. A *normalizing flow* is a vector-valued invertible mapping $\mathbf{f}(\mathbf{x}) : \mathbb{R}^D \to \mathbb{R}^D$ that normalizes the distribution of $\mathbf{x}$ to a "standard" distribution (or called *base distribution*). This distribution is usually taken to be an isotropic Gaussian or other ones that are easy to sample from and whose density is easy to evaluate. Let $\mathbf{z} = \mathbf{f}(\mathbf{x})$ with probability density function $q(\mathbf{z})$. With the change-of-variable formula, we can express the density of the $\mathbf{x}$, $p(\mathbf{x})$, by:

$$\log p(\mathbf{x}) = \log q(\mathbf{f}(\mathbf{x})) + \log |\det \nabla_{\mathbf{x}} \mathbf{f}(\mathbf{x})|. \tag{1}$$

In practical uses, the Jacobian determinant in (1) needs be easy to compute, so that the density $p(\mathbf{x})$ can be evaluated. Moreover, as a generative model, the invertibility of $\mathbf{f}$ allows drawing new instances $\mathbf{x} = \mathbf{f}^{-1}(\mathbf{z})$ through sampling the base distribution. One example of such $\mathbf{f}$ is the masked autoregressive flow (Papamakarios et al., 2017), which yields $\mathbf{z} = [z_1, \ldots, z_D]$ from $\mathbf{x} = [x_1, \ldots, x_D]$ through

$$z_i = (x_i - \mu_i(\mathbf{x}_{1:i-1})) \exp(\alpha_i(\mathbf{x}_{1:i-1})), \tag{2}$$

where $\mu_i$ and $\alpha_i$ are neural networks such as the multilayer perceptron.

A flow may be augmented with conditional information $\mathbf{h} \in \mathbb{R}^d$ with a possibly different dimension. Such a flow is a *conditional flow* and is denoted by $\mathbf{f} : \mathbb{R}^D \times \mathbb{R}^d \to \mathbb{R}^D$. The log-density of $\mathbf{x}$ conditioned on $\mathbf{h}$ admits the following formula:

$$\log p(\mathbf{x}|\mathbf{h}) = \log q(\mathbf{f}(\mathbf{x}; \mathbf{h})) + \log |\det \nabla_{\mathbf{x}} \mathbf{f}(\mathbf{x}; \mathbf{h})|. \tag{3}$$

We now consider a *normalizing flow for time series*. Let $\mathbf{X} = [\mathbf{x}_1, \mathbf{x}_2, \ldots, \mathbf{x}_T]$ denote a time series of length $T$, where $\mathbf{x}_t \in \mathbb{R}^D$. Through successive conditioning, the density of the time series can be written as:

$$p(\mathbf{X}) = p(\mathbf{x}_1)p(\mathbf{x}_2|\mathbf{x}_{<2}) \cdots p(\mathbf{x}_T|\mathbf{x}_{<T}), \tag{4}$$

where $\mathbf{x}_{<t}$ denotes all variables before time $t$. When the conditional probabilities are parameterized, Rasul et al. (2021) propose to model each $p(\mathbf{x}_t|\mathbf{x}_{<t})$ as $p(\mathbf{x}_t|\mathbf{h}_{t-1})$, where $\mathbf{h}_{t-1}$ summarizes the

past information $\mathbf{x}_{<t}$. For example, $\mathbf{h}_{t-1}$ is the hidden state of a recurrent neural network before accepting input $\mathbf{x}_t$. Then, a conditional normalizing flow can be applied to evaluate each $p(\mathbf{x}_t|\mathbf{h}_{t-1})$.

## 3.2 BAYESIAN NETWORKS

Let $X^i$ denote a general random variable, either scalar valued, vector valued, or even matrix valued. A *Bayesian network* of $n$ variables $(X^1, \ldots, X^n)$ is a directed acyclic graph of the variables as nodes. Let $\mathbf{A}$ denote the weighted adjacency matrix of the graph, where $\mathbf{A}_{ij} \neq 0$ if $X^j$ is the parent of $X^i$. A Bayesian network describes the conditional independence among variables. Specifically, a node $X^i$ is conditionally independent of its non-descendents given its parents. In other words, the density of the joint distribution of $(X^1, \ldots, X^n)$ is

$$p(X^1, \ldots, X^n) = \prod_{i=1}^{n} p(X^i | \operatorname{pa}(X^i)), \tag{5}$$

where $\operatorname{pa}(X^i) = \{X^j : \mathbf{A}_{ij} \neq 0\}$ denotes the set of parents of $X^i$.

## 4 PROBLEM STATEMENT

In this paper, we focus on unsupervised anomaly detection with multiple time series. The training set $\mathcal{D}$ consists of only unlabeled instances and we assume that the majority of them are not anomalies. Each instance $\mathcal{X} \in \mathcal{D}$ contains $n$ constituent series with $D$ attributes and of length $T$; i.e., $\mathcal{X} = (\mathbf{X}^1, \mathbf{X}^2, \ldots, \mathbf{X}^n)$ where $\mathbf{X}^i \in \mathbb{R}^{T \times D}$. We use a Bayesian network (DAG) to model the relational structure of the constituent series $\mathbf{X}^i$ and augment a normalizing flow to compute the density of $\mathcal{X}$ through a factorization in the form (5). Let $\mathbf{A} \in \mathbb{R}^{n \times n}$ be the adjacency matrix of the DAG and let $\mathcal{F} : (\mathcal{X}, \mathbf{A}) \to \mathcal{Z}$ denote the augmented flow. Because anomaly points tend to have low densities, we propose to conduct unsupervised anomaly detection by evaluating the density of a multiple time series computed through the augmented flow. The problem is formulated as the following.

**Problem 1.** *Given a training set $\mathcal{D} = \{\mathcal{X}_i\}_{i=1}^{|\mathcal{D}|}$ of multiple time series, we aim to simultaneously learn the adjacency matrix $\mathbf{A}$ of the Bayesian Network that represents the conditional dependencies among the constituent series, as well as the correspondingly graph-augmented normalizing flow $\mathcal{F} : (\mathcal{X}, \mathbf{A}) \to \mathcal{Z}$, which is used to estimate the density of an instance $\mathcal{X}$. Here, $\mathcal{Z}$ is a random variable with a "simple" distribution, such as the anisotropic Gaussian.*

## 5 METHOD

In this section, we materialize the graph-augmented normalizing flow $\mathcal{F} : (\mathcal{X}, \mathbf{A}) \to \mathcal{Z}$ introduced in the problem statement and use it to compute the density of a multiple time series $\mathcal{X}$. The central idea is factorization: we factorize $p(\mathcal{X})$ along the series dimension by using a Bayesian network and then factorize along the temporal dimension by using conditional normalizing flows. Then, we employ a novel graph-based dependency encoder to parameterize the conditional probabilities resulting from the factorization. The DAG used for factorization is a discrete object and is usually intractable to learn; however, the discrete structure is reflected in the dependency encoder through a graph adjacency matrix $\mathbf{A}$ that is differentiable. Moreover, the requirement that $\mathbf{A}$ must correspond to a DAG can be expressed as a differentiable equation. Hence, one can jointly optimize $\mathbf{A}$ and the flow components by using gradient based optimization. Once $\mathcal{F}$ is learned, the density $p(\mathcal{X})$ is straightforwardly evaluated for anomaly detection. An illustration of the framework GANF is shown in Figure 1.

### 5.1 FACTORIZATION

Figure 1 shows a toy example of a Bayesian network as a DAG. Based on (5), the density of a multiple time series $\mathcal{X} = (\mathbf{X}^1, \mathbf{X}^2, \ldots, \mathbf{X}^n)$ can be computed as the product of $p(\mathbf{X}^i | \operatorname{pa}(\mathbf{X}^i))$ for all nodes, where recall that $\operatorname{pa}(\mathbf{X}^i)$ denotes the set of parents of $\mathbf{X}^i$. Then, following Rasul et al. (2021), we further factorize each conditional density along the temporal dimension. Specifically, for a time step $t$, $\mathbf{x}_t^i$ depends on its past history as well as its parents in the DAG. We write

$$p(\mathcal{X}) = \prod_{i=1}^{n} p(\mathbf{X}^i | \operatorname{pa}(\mathbf{X}^i)) = \prod_{i=1}^{n} \prod_{t=1}^{T} p(\mathbf{x}_t^i | \operatorname{pa}(\mathbf{x}^i)_{1:t}, \mathbf{x}_{1:t-1}^i), \tag{6}$$

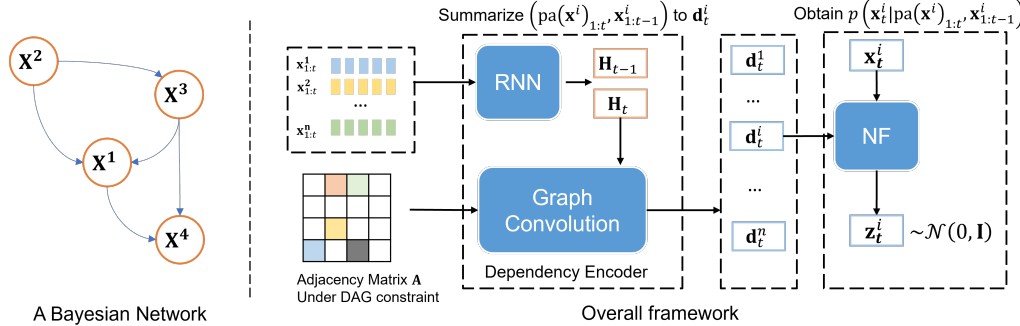

Figure 1: Illustration of a Bayesian network and the proposed framework GANF.

where $\mathbf{x}_{1:t-1}^i$ denotes the history of node $i$ before time $t$, and $\mathrm{pa}(\mathbf{x}^i)_{1:t} = \{\mathbf{x}_{1:t}^j : \mathbf{A}_{ij} \neq 0\}$. In the next subsection, we will parameterize each conditional density $p(\mathbf{x}_t^i \mid \mathrm{pa}(\mathbf{x}^i)_{1:t}, \mathbf{x}_{1:t-1}^i)$ by using a graph-based dependency encoder. Note that so far the factorization (6) has been based on the discrete structure of the Bayesian network. The dependency encoder we introduce next, however, uses the adjacency matrix $\mathbf{A}$ in a differentiable manner, which is sufficient to ensure that $\mathbf{x}_t^i$ will not depend on nodes other than its parents and itself.

## 5.2 NEURAL NETWORK PARAMETERIZATION

According to Sec. 3.1, conditional densities $p(\mathbf{x}_t^i \mid \mathrm{pa}(\mathbf{x}^i)_{1:t}, \mathbf{x}_{1:t-1}^i)$ can be learned by using conditional normalizing flows. However, the conditional information $\mathrm{pa}(\mathbf{x}^i)_{1:t}$ and $\mathbf{x}_{1:t-1}^i$ cannot be directly used for parameterization, because its size is not fixed. Therefore, as is illustrated in Figure 1, we design a graph-based dependency encoder to summarize the conditional information into a fixed length vector $\mathbf{d}_t^i \in \mathbb{R}^d$. Then, a conditional normalizing flow is used to evaluate $p(\mathbf{x}_t^i \mid \mathbf{d}_t^i)$, which is equivalent to $p(\mathbf{x}_t^i \mid \mathrm{pa}(\mathbf{x}^i)_{1:t}, \mathbf{x}_{1:t-1}^i)$.

**Dependency Encoder.** Since the history has an arbitrary length, we first employ a recurrent neural network (RNN) to map multiple time steps to a vector of fixed length. For a time series $\mathbf{x}_{1:t}^i$, the recurrent model abstracts it into a hidden state $\mathbf{h}_t^i \in \mathbb{R}^d$ through the following recurrence

$$\mathbf{h}_t^i = \mathrm{RNN}(\mathbf{x}_t^i, \mathbf{h}_{t-1}^i), \tag{7}$$

where $\mathbf{h}_t^i$ summarizes the time series up to step $t$. The RNN can be any sequential model, such as the LSTM (Hochreiter & Schmidhuber, 1997) and in a broad sense a transformer (Vaswani et al., 2017). We let the RNN parameters be shared across all nodes in the DAG to avoid overfitting and to reduce computational costs.

With (7), the conditional information of $\mathbf{x}_t^i$ is all summarized in $\{\mathbf{h}_t^j : \mathbf{A}_{ij} \neq 0\} \cup \{\mathbf{h}_{t-1}^i\}$. Inspired by the success of GCN (Kipf & Welling, 2016) in node representation learning through neighborhood aggregation, we design a graph convolution layer to aggregate hidden states of the parents for dependency encoding. This layer produces dependency representations $\mathbf{D}_t = (\mathbf{d}_t^1, \dots, \mathbf{d}_t^n)$ for all constituent series at time $t$:

$$\mathbf{D}_t = \mathrm{ReLU}(\mathbf{A}\mathbf{H}_t\mathbf{W}_1 + \mathbf{H}_{t-1}\mathbf{W}_2) \cdot \mathbf{W}_3, \tag{8}$$

where $\mathbf{H}_t = (\mathbf{h}_t^1, \dots, \mathbf{h}_t^n)$ contains all the hidden states at time $t$. Here, $\mathbf{W}_1 \in \mathbb{R}^{d \times d}$ and $\mathbf{W}_2 \in \mathbb{R}^{d \times d}$ are parameters to transform the aggregated representations of the parents and the node's historical information, respectively; while $\mathbf{W}_3 \in \mathbb{R}^{d \times d}$ is an additional transformation to improve the dependency representation.

**Density Estimation.** With the dependency encoder, we obtain the representations $\mathbf{d}_t^i$ of the conditional information. Then, a normalizing flow $\mathbf{f} : \mathbb{R}^D \times \mathbb{R}^d \to \mathbb{R}^D$ conditioned on $\mathbf{d}_t^i$ is applied to model each $p(\mathbf{x}_t^i \mid \mathrm{pa}(\mathbf{x}^i)_{1:t}, \mathbf{x}_{1:t-1}^i)$. Similar to the computation of the hidden states, the parameters of the conditional flow are also shared among nodes, to avoid overfitting. Based on (3), the conditional density of $\mathbf{x}_t^i$ can be written as:

$$\log p(\mathbf{x}_t^i \mid \mathrm{pa}(\mathbf{x}^i)_{1:t}, \mathbf{x}_{1:t-1}^i) = \log p(\mathbf{x}_t^i \mid \mathbf{d}_t^i) = \log q(\mathbf{f}(\mathbf{x}_t^i; \mathbf{d}_t^i)) + \log |\det \nabla_{\mathbf{x}_t^i} \mathbf{f}(\mathbf{x}_t^i; \mathbf{d}_t^i)|, \tag{9}$$

where $q(\mathbf{z})$ is chosen to be the standard normal $\mathcal{N}(\mathbf{z}|\mathbf{0}, \mathbf{I})$ with $\mathbf{z} \in \mathbb{R}^D$. The conditional flow $\mathbf{f}$ can be any effective one proposed by the literature, such as RealNVP (Dinh et al., 2016) and

MAF (Papamakarios et al., 2017). Combining (9) and (6), we obtain the log-density of a multiple time series $\mathcal{X}$:

$$\log p(\mathcal{X}) = \sum_{i=1}^{n} \sum_{t=1}^{T} \Big[ \log q(\mathbf{f}(\mathbf{x}_t^i; \mathbf{d}_t^i)) + \log |\det \nabla_{\mathbf{x}_t^i} \mathbf{f}(\mathbf{x}_t^i; \mathbf{d}_t^i)| \Big]. \tag{10}$$

**Anomaly Measure.** Because anomalies deviate significantly from the majority of the data instances, we hypothesize that their densities are low. Thus, we use the density computed by (10) as the anomaly measure, where a lower density indicates a more likely anomaly. Apart from evaluating the density for the entire $\mathcal{X}$, the computation also produces conditional densities $\log p(\mathbf{X}^i | \operatorname{pa}(\mathbf{X}^i)) = \sum_{t=1}^{T} \log p(\mathbf{x}_t^i | \mathbf{d}_t^i)$ for each constituent series $\mathbf{X}^i$. We use this conditional density as the anomaly measure for constituent series. A low density $p(\mathcal{X})$ is caused by one or a few low conditional densities $p(\mathbf{X}^i | \operatorname{pa}(\mathbf{X}^i))$ in the Bayesian network, suggesting that abnormal behaviors could be traced to individual series.

### 5.3 Joint Training

Learning a Bayesian network is a challenging combinatorial problem, due to the intractable search space superexponential in the number of nodes. A recent work by Zheng et al. (2018) proposes the equation $\operatorname{tr}(e^{\mathbf{A} \circ \mathbf{A}}) = n$ that characterizes the acyclicity of the corresponding graph of $\mathbf{A}$, where $e$ is matrix exponential and $\circ$ denotes element-wise multiplication. We will impose this equation as a constraint in the training of GANF.

**Training Objective.** Following the training of a usual normalizing flow, the joint density (likelihood) of the observed data is the training objective, which is equivalent to the Kullback–Leibler divergence between the true distribution of data and the flow recovered distribution. Together with the DAG constraint, the optimization problem reads

$$\min_{\mathbf{A}, \boldsymbol{\theta}} \quad \mathcal{L}(\mathbf{A}, \boldsymbol{\theta}) = \frac{1}{|\mathcal{D}|} \sum_{i=1}^{|\mathcal{D}|} - \log p(\mathcal{X}_i), \tag{11}$$

$$\text{s.t.} \quad h(\mathbf{A}) = \operatorname{tr}(e^{\mathbf{A} \circ \mathbf{A}}) - n = 0,$$

where $\boldsymbol{\theta}$ contains all neural network parameters, including those of the dependency encoder and the normalizing flow. Here, the DAG constraint $h(\mathbf{A})$ admits an easy-to-evaluate gradient $\nabla h(\mathbf{A}) = (e^{\mathbf{A} \circ \mathbf{A}})^T \circ 2\mathbf{A}$, which allows a gradient based optimizer to solve (11).

**Training Algorithm.** Problem (11) is a nonlinear equality-constrained optimization. Such problems are extensively studied and the augmented Lagrangian method (Bertsekas, 1999; Yu et al., 2019) is one of the most widely used approaches. The augmented Lagrangian is defined as

$$\mathcal{L}_c = \mathcal{L}(\mathbf{A}, \boldsymbol{\theta}) + \lambda h(\mathbf{A}) + \frac{c}{2} |h(\mathbf{A})|^2, \tag{12}$$

where $\lambda$ and $c$ denote the Lagrange multiplier and the penalty parameter, respectively. The general idea of the method is to gradually increase the penalty parameter to ensure that the constraint is eventually satisfied. Over iterations, $\lambda$ as a dual variable will converge to the Lagrangian multiplier of (11). The update rule at the $k$th iteration reads the following:

$$\mathbf{A}^k, \boldsymbol{\theta}^k = \arg \min_{\mathbf{A}, \boldsymbol{\theta}} \mathcal{L}_{c^k}; \quad \lambda^{k+1} = \lambda^k + c^k h(\mathbf{A}^k); \quad c^{k+1} = \begin{cases} \eta c^k & \text{if } |h(\mathbf{A}^k)| > \gamma |h(\mathbf{A}^{k-1})|; \\ c^k & \text{else}, \end{cases} \tag{13}$$

where $\eta \in (1, +\infty)$ and $\gamma \in (0, 1)$ are hyperparameters to be tuned. We set $\eta$ and $\gamma$ as 10 and 0.5, respectively. The subproblem of optimizing $\mathbf{A}$ and $\boldsymbol{\theta}$ can be solved by using the Adam optimizer (Kingma & Ba, 2014). The training algorithm is summarized in Appendix A.

## 6 Experiments

In this section, we conduct a comprehensive set of experiments to validate the effectiveness of the proposed GANF framework. In particular, they are designed to answer the following questions:

- **Q1:** Can GANF accurately detect anomalies and estimate densities?

- **Q2:** Does the proposed graph structure learning help? Is the framework sufficiently flexible to include various normalizing flow backbones?

- **Q3:** What can one observe for a dataset spanning a long time? E.g., does the graph pattern change?

### 6.1 SETTINGS

**Datasets.** To evaluate the effectiveness of GANF for anomaly detection and density estimation, we conduct experiments on two power grid datasets, one water system dataset, and one traffic dataset.

- **PMU-B** and **PMU-C**: These two datasets correspond to two separate interconnects of the U.S. power grid, containing time series recorded by 38 and 132 phasor measurement units (PMUs), respectively. We process one-year data at the frequency of one second to form a ten-month training set, one-month validation set, and one-month test set. Each multiple time series is obtained by shifting a one-minute window. Additionally, to investigate distribution drift, we shift a one-month window to obtain multiple training/validation/test sets (12 in total, because of availability of two-year data). Sparse grid events (anomalies) labeled by domain experts exist for evaluation; but note that the labels are both noisy and incomplete. These datasets are proprietary.

- **SWaT**: We also use a public dataset for evaluation. The Secure Water Treatment (SWaT) dataset originates from an operational water treatment test-bed coordinated with Singapore's Public Utility Board (Goh et al., 2016). The data collects 51 sensor recordings lasting four days, at the frequency of one second. A total of 36 attacks were conducted, resulting in approximately 11% time steps as anomaly ground truths. We use a sliding window of 60 seconds to construct series data and perform a 60/20/20 chronological split for training, validation, and testing, respectively.

- **METR-LA**: This dataset is also public; it contains speed records of 207 sensors deployed on the highways of Los Angles, CA (Li et al., 2018b). No anomaly labels exist however and we use this dataset for exploratory analysis only. Results are deferred to Appendix E.

**Evaluation metrics (under noisy labels).** For SWaT, which offers reliable ground truths, we use the standard ROC and AUC metrics for evaluation. For the two PMU datasets, however, the resolution of the time series and the granularity of the events result in rather noisy ground truths. Hence, we adapt ROC for noisy labels. We smooth the time point of a "ground truth" event (anomaly) by introducing probabilities to the label. Specifically, the probability that a multiple time series starting at time $t$ is a ground truth anomaly is $\max_i\{\exp(-\frac{(t-t_i)^2}{\sigma^2})\}$, where $t_i$ is the starting time of the $i$th labeled anomaly. Then, when computing the confusion matrix, we sum probabilities rather than counting 0/1s. The smoothing window $\sigma$ is chosen to be 6 time steps.

**Baselines.** We compare with the following representative, state-of-the-art deep methods.

- **EncDecAD** (Malhotra et al., 2016): In this method, an autoencoder based on LSTM is trained. The reconstruction error is used as the anomaly measure.

- **DeepSVDD** (Ruff et al., 2018): This method minimizes the volume of a hypersphere that encloses the representations of data. Samples distant from the hypersphere center are considered anomalies.

- **ALOCC** (Sabokrou et al., 2020): In this GAN-based method, the generator learns to reconstruct normal instances, while the discriminator works as an anomaly detector.

- **DROCC** (Goyal et al., 2020): This method performs adversarial training to learn robust representations of data and identifies anomalies.

- **DeepSAD** (Ruff et al., 2020): This method extends DeepSVDD with a semi-supervised loss term for training. We use noisy labels as supervision.

To apply these baselines on multiple time series, we concatenate the constituent series along the attribute dimension (resulting in high-dimensional series) and use LSTM or CNN as the backbones. On the other hand, for the proposed method, we use LSTM as the RNN model and MAF as the normalizing flow. See Appendix C for more implementation details.

### 6.2 PERFORMANCE OF ANOMALY DETECTION AND DENSITY ESTIMATION

Table 1: AUC-ROC (%) of anomaly detection.

| Dataset | EncDecAD | DeepSVDD | ALOCC | DROCC | DeepSAD | GANF |
|---------|----------|----------|-------|-------|---------|------|
| PMU-B | 55.6±1.8 | 55.6±3.3 | 62.9±2.2 | 58.6±3.0 | 63.7±0.9 | **67.5**±0.8 |
| PMU-C | 53.7±0.5 | 56.9±0.9 | 60.9±1.3 | 61.9±2.7 | 60.1±1.4 | **70.6**±3.3 |
| SWaT | 76.5±0.7 | 68.8±2.0 | 75.4±2.3 | 73.3±1.6 | 75.4±1.2 | **79.6**±0.9 |

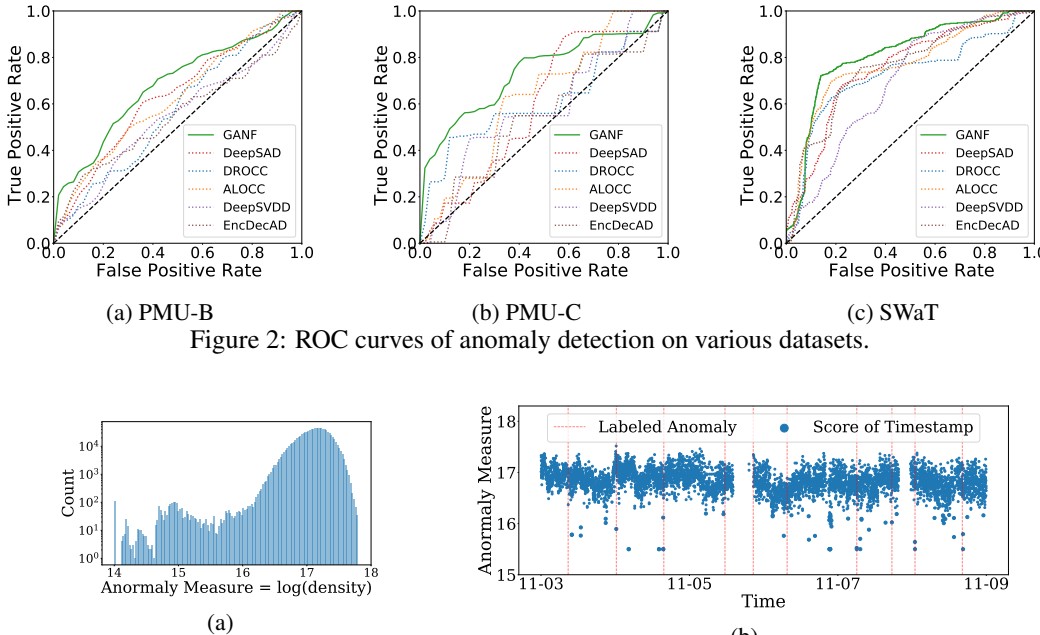

(a) PMU-B                        (b) PMU-C                        (c) SWaT

Figure 2: ROC curves of anomaly detection on various datasets.

Figure 3: Qualitative evaluation of GANF on PMU-C. (a) Distribution of log-densities on the test set (note in log scale). (b) Anomaly detection results for a week in the test set.

To answer **Q1**, we evaluate quantitatively and qualitatively on datasets with labels.

**Anomaly detection.** We compare GANF with the aforementioned baselines in Table 1, where standard deviations of the AUC scores are additionally reported through five random repetitions of model training. The table suggests an overwhelmingly high AUC score achieved by GANF. Observations follow. (**i**) GANF outperforms generative model-based methods (EncDecAD and ALOCC). Being a generative model as well, normalizing flows augmented with a graph structure leverage the interdependencies of constituent series more effectively, leading to a substantial improvement in detection. (**ii**) GANF significantly outperforms deep one-class models (DeepSVDD and DROCC), corroborating the appeal of using densities for detection. (**iii**) GANF also performs better than the semi-supervised method DeepSAD, probably because such methods rely on high quality labels for supervision (especially in the case of label scarcity) and they are less effective facing noisy labels.

Besides a single score, we also plot the ROC curve in Figure 2. One sees that the curve of GANF dominates those of others. This behavior is generally more salient in the low false-alarm regime.

**Density estimation.** We investigate the densities estimated by GANF, shown in Figure 3. Distributions of the log-densities in the test set are shown in Figure 3a. We use log-density as the anomaly measure; the lower the more likely. Note that the vertical axis is in the log-scale. One sees that a log-density of 16 approximately separates the majority normal instances from the minority anomalies. To cross-verify that the instances with low densities are suspiciously anomalous, we investigate Figure 3b, which is a temporal plot of log-densities for a week, overlaid with given labels. From this plot, one sees that the noisily labeled series generally have low densities or are near a low density time step. Additionally, GANF discovers a few suspicious time steps with low densities undetected earlier. These new discoveries raise interest to power system experts for analysis and archiving.

## 6.3 ABLATION STUDY

Table 2: Performance of variants of the proposed method.

| Dataset | Metrics | GANF\G | GANF\D | GANF\T | GANF$_{RNVP}$ | GANF |
|---------|---------|--------|--------|--------|---------------|------|
| PMU-B | AUC-ROC | 0.641 | 0.643 | 0.653 | 0.661 | **0.678** |
|  | Log-Density | 15.31 | 8.70 | 15.09 | 15.90 | **16.22** |
| PMU-C | AUC-ROC | 0.630 | 0.544 | 0.688 | 0.703 | **0.705** |
|  | Log-Density | 15.55 | 8.94 | 15.70 | **17.06** | 16.98 |

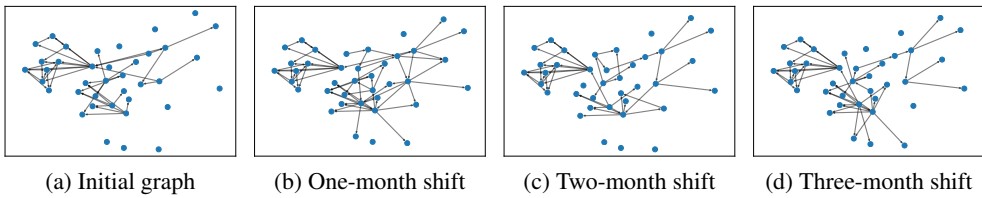

(a) Initial graph     (b) One-month shift     (c) Two-month shift     (d) Three-month shift

Figure 4: Evolution of the learned DAG on PMU-B over time.

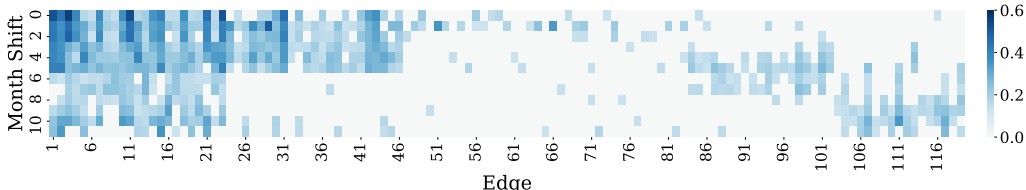

Figure 5: Evolution of edge weights in the DAG learned by GANF over time (PMU-B).

To answer **Q2**, we conduct an ablation study (including varying architecture components) to investigate impacts of DAG structure learning and the flexibility of the GANF framework. To investigate the power of modeling pairwise relationship, we train a variant GANF\G that factorizes $p(\mathcal{X}) = \prod_{i=1}^{n} p(\mathbf{X}^i)$; i.e., assuming independence among constituent series. To investigate the effectiveness of graph structure learning, we train a variant GANF\D that decomposes the joint density as $p(\mathcal{X}) = \prod_{i=1}^{n} p(\mathbf{X}^i|\mathbf{X}^{<i})$; i.e., a full decomposition without a DAG. It is equivalent to concatenating the series along the attribute dimension and running MAF on the resulting series. To verify the contribution of joint training of $\mathbf{A}$ and $\boldsymbol{\theta}$, we train a variant GANF\T where $\mathbf{A}$ is separately learned by using NOTEARS (Zheng et al., 2018). To prove the flexibility of GANF, we replace the MAF-based normalizing flow by RealNVP, denoted as GANF$_{\text{RNVP}}$.

Results are presented in Table 2. Apart from AUC-ROC, the log-density is also reported. Observations follow. (**i**) GANF significantly outperforms GANF\G and GANF\D, corroborating the importance of interdependency modeling among constituent series. Note that GANF\D results in particularly poor performance in general, likely because the high dimensional input (resulting from concatenating too many series) impedes the learning of normalizing flows. (**iii**) GANF\T is slightly better than GANF\G, because of the presence of relational modeling, but it cannot match the performance of GANF$_{\text{RNVP}}$ and GANF that jointly train the DAG and the flow. (**ii**) These latter two models are the best for both datasets and both metrics. MAF works more often better than RealNVP.

## 6.4 Evolution of the DAG Structure

To answer **Q3**, we investigate how the learned DAG evolves by shifting the train/validation/test sets month by month. The graphs within the first three-month shiftings are shown in Figure 4 and more can be found in Appendix F. In addition to the graph structure, we plot in Figure 5 the learned edge weights over time, one column per edge. The appearance and disappearance of edges demonstrate changes of the conditional independence structure among constituent series over time, suggesting data distribution drift (i.e., a change of internal data generation mechanism). It is interesting to observe the seasonal effect. The columns (edges) in Figure 5 can be loosely grouped in three clusters: those persisting the entire year, those appearing in the first half of the year, and those existing more briefly (e.g., within a season). Such a pattern plausibly correlates with electricity consumption, which is also seasonal. Were spatial information of the PMUs known, these identified DAGs would help mapping the seasonal patterns to geography and help planning a more resilient grid.

## 7 Conclusions

In this paper, we present a graph-augmented normalizing flow GANF for anomaly detection of multiple time series. The graph is materialized as a Bayesian network, which models the conditional dependencies among constituent time series. A graph-based dependency decoder is designed to summarize the conditional information needed by the normalizing flow that calculates series density. Anomalies are detected through identifying instances with low density. Extensive experiments on real-world datasets demonstrate the effectiveness of the framework. Ablation studies confirm the contribution of the learned graph structure in anomaly detection. Additionally, we investigate the evolution of the graph and offer insights of distribution drift over time.

ACKNOWLEDGMENT AND DISCLAIMER

This material is based upon work supported by the Department of Energy under Award Number(s) DE-OE0000910. This report was prepared as an account of work sponsored by an agency of the United States Government. Neither the United States Government nor any agency thereof, nor any of their employees, makes any warranty, express or implied, or assumes any legal liability or responsibility for the accuracy, completeness, or usefulness of any information, apparatus, product, or process disclosed, or represents that its use would not infringe privately owned rights. Reference herein to any specific commercial product, process, or service by trade name, trademark, manufacturer, or otherwise does not necessarily constitute or imply its endorsement, recommendation, or favoring by the United States Government or any agency thereof. The views and opinions of authors expressed herein do not necessarily state or reflect those of the United States Government or any agency thereof.

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

## A  TRAINING ALGORITHM

We summarize the training method in Algorithm 1.

---

**Algorithm 1** Training Algorithm of GANF

---

**Input:** Training set $\mathcal{D}$, hyperparameters $\eta$ and $\gamma$
**Output:** GANF $\mathcal{F}$ and adjacency matrix $\mathbf{A}$ of the DAG
 1: Initialize $c \leftarrow 0$ and initialize $\lambda$ randomly
 2: **for** $k = 0, 1, 2, \ldots$ **do**
 3:     Compute $\mathbf{A}^k$ and $\boldsymbol{\theta}^k$ as a minimizer of (12) by using the Adam optimizer, where the loss
        $\mathcal{L}$ and the constraint $h$ are defined in (11), the log-density $\log p(\mathcal{X})$ is defined in (10), the
        dependency representation $\mathbf{d}_t^i$ is defined in (8), the hidden state $\mathbf{h}_t^i$ is defined in (7), and the
        conditional flow $\mathbf{f}$ is RealNVP or MAF
 4:     Update Lagrange multiplier $\lambda \leftarrow \lambda + ch(\mathbf{A}^k)$
 5:     **if** $k > 0$ and $|h(\mathbf{A}^k)| > \gamma |h(\mathbf{A}^{k-1})|$ **then**
 6:         $c \leftarrow \eta c$
 7:     **end if**
 8:     **if** $h(\mathbf{A}^k) == 0$ **then**
 9:         break
10:     **end if**
11: **end for**
12: **return** $\mathbf{A}$ and $\mathcal{F}$ (including $\mathbf{f}$, the neural network (8), and the RNN (7))

---

## B  CODE

Code is available at `https://github.com/EnyanDai/GANF`.

## C  ADDITIONAL DETAILS OF EXPERIMENT SETTINGS

### C.1  IMPLEMENTATION DETAILS OF GANF.

An LSTM is used as the RNN model in the dependency encoder. For normalizing flows, we use MAF with six flow blocks. All hidden dimensions as set as 32. The initial learning rate is set as 0.001 for the adjacency matrix $\mathbf{A}$ and the model parameters $\boldsymbol{\theta}$. Learning rate decay is 0.1. To avoid gradient explosion, we clip the gradients whose values are larger than 1.0.

For hyperparameter tuning, we select the hyperparameters that yield the highest log-density on the validation set. Specifically, we conduct grid search by varying the number of normalizing flow blocks from $\{1, 2, 4, 6, 8\}$, the learning rate from $\{0.003, 0.001, 0.0003, 0.0001\}$, and the hidden dimension from $\{16, 32, 64, 128\}$.

### C.2  IMPLEMENTATION DETAILS OF BASELINES.

- **EncDecAD** (Malhotra et al., 2016): This method is applied for anomaly detection on time series. Thus, we concatenate constituent series along the attribute dimension and adopt the code released by the authors in `https://github.com/chickenbestlover/RNN-Time-series-Anomaly-Detection`.

- **DeepSVDD** (Ruff et al., 2018): To handle time series data, we replace the backbone to an LSTM based on the official implementation `https://github.com/lukasruff/Deep-SVDD-PyTorch`.

- **ALOCC** (Sabokrou et al., 2020): We use the official implementation released by the authors in `https://github.com/khalooei/ALOCC-CVPR2018`. We replace the two-dimensional convolution to one-dimensional convolution to build a GAN for time series data.

- **DROCC** (Goyal et al., 2020): Similar to other baselines, this method is proposed for tabular data and image data. We replace the backbone to LSTM to deal with multiple time series by revising the encoder in `https://github.com/microsoft/EdgeML/tree/master/pytorch`.

- **DeepSAD** (Ruff et al., 2020): This is a semi-supervised approach, which requires labeling. We utilize the noisy labels in PMU-B and PMU-C as supervision. We use LSTM as the

backbone, based on the official implementation in `https://github.com/lukasruff/Deep-SAD-PyTorch`.

All hyperparameters of the baselines are tuned based on the validation set to make fair comparisons.

## D  TIME COMPLEXITY ANALYSIS

The GANF framework involves Bayesian network structure learning, which is known to be highly challenging, owing to the intractable search space superexponential in the number of graph nodes. In this work, we formulate a continuous optimization of the graph structure, so that the training of GANF is more scalable. In what follows, we analyze the time complexity.

Recall that each instance $\mathcal{X}$ of the multiple time series dataset contains $n$ constituent series with $D$ attributes and of length $T$; i.e., $\mathcal{X} = (\mathbf{X}^1, \mathbf{X}^2, \dots, \mathbf{X}^n)$ where $\mathbf{X}^i \in \mathbb{R}^{T \times D}$. In the evaluation of the model, the dominant costs appear in running the dependency encoder and the normalizing flow. For the dependency encoder, an RNN is first deployed to map the multiple time series to hidden vectors; the time complexity is $O(nTD)$. Then, graph convolution is conducted to obtain dependency vectors; the convolution cost is $O(n^2T)$. For the normalizing flow module, the time complexity is $O(nTD)$. Therefore, the time complexity of computing log-density of one instance is $O(nT(D+n))$. If we use a batch size $B$ for training, the time cost of calculating the augmented Lagrangian $\mathcal{L}(\mathbf{A}, \boldsymbol{\theta})$ in (12) is $O(nBT(D+n))$. Additionally, the time cost of calculating the constraint $h(\mathbf{A})$ is $O(n^3)$. Thus, the overall time complexity of one training iteration is $O(n(BTD + BTn + n^2))$.

## E  RESULTS FOR METR-LA

METR-LA contains speed records of 207 sensors deployed on the highways of Los Angles, CA (Li et al., 2018b). The records are in four months at the frequency of five minutes. We shift a one-hour window to obtain multiple time series. The first three months are used for training and the last month is split in halves for validation and testing. No anomaly labels exist however and we use this dataset for exploratory analysis only.

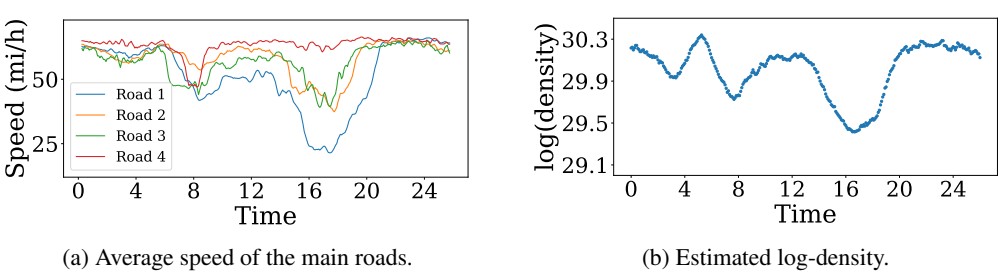

(a) Average speed of the main roads.  (b) Estimated log-density.

Figure 6: Density estimation for METR-LA.

Figure 6a shows the traffic speed on four main highways on June 13, 2012. Each speed is the average over all sensors on the same highway. We observe that despite spatial proximity, the speeds vary significantly around 4PM (rush hour) but they are unanimously high around 5AM and 8PM–12AM. The estimated densities, shown in Figure 6b, tracks this pattern rather closely, with rush hours corresponding to low density and night traffics corresponding to high density. Note the nature of traffic: speed varies smoothly on the macroscopic level and hence does density, too. Such a phenomenon is in striking contrast to power systems where events are rare and abrupt.

## F    ADDITIONAL ANOMALY DETECTION RESULTS ON PMU DATASETS

See Figure 7 and Figure 8 for additional anomaly detection results on the test sets of PMU-C and PMU-B, respectively. The observations are rather similar to those of Figure 3b.

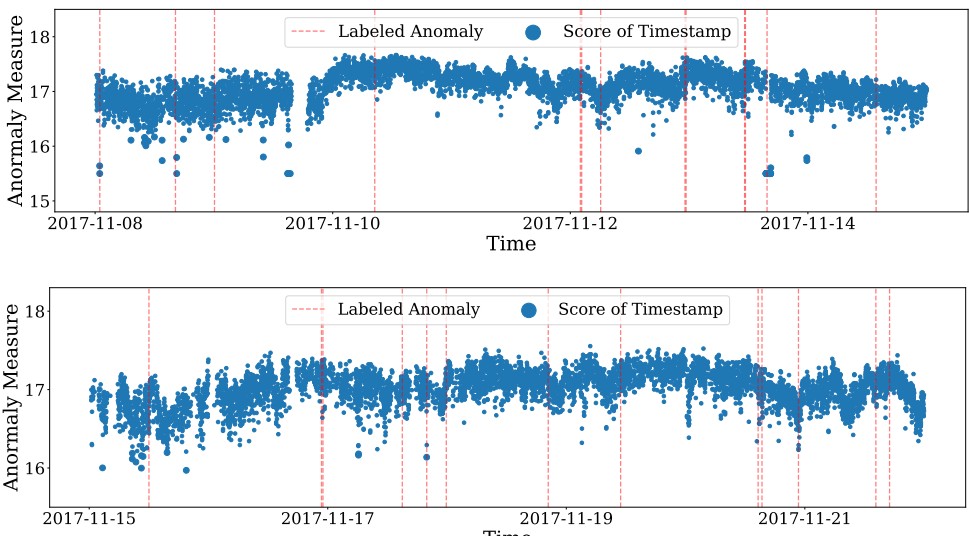

Figure 7: Additional anomaly detection results on the test set of PMU-C.

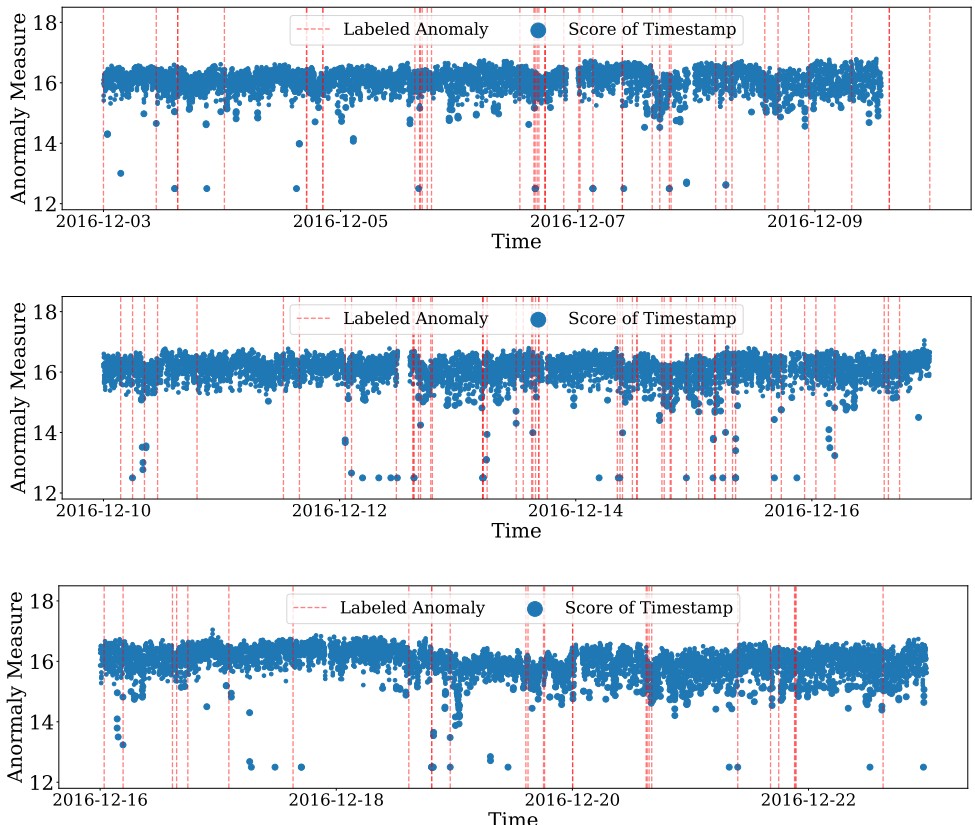

Figure 8: Anomaly detection results on the test set of PMU-B.

# G ADDITIONAL RESULTS FOR DAG EVOLUTION

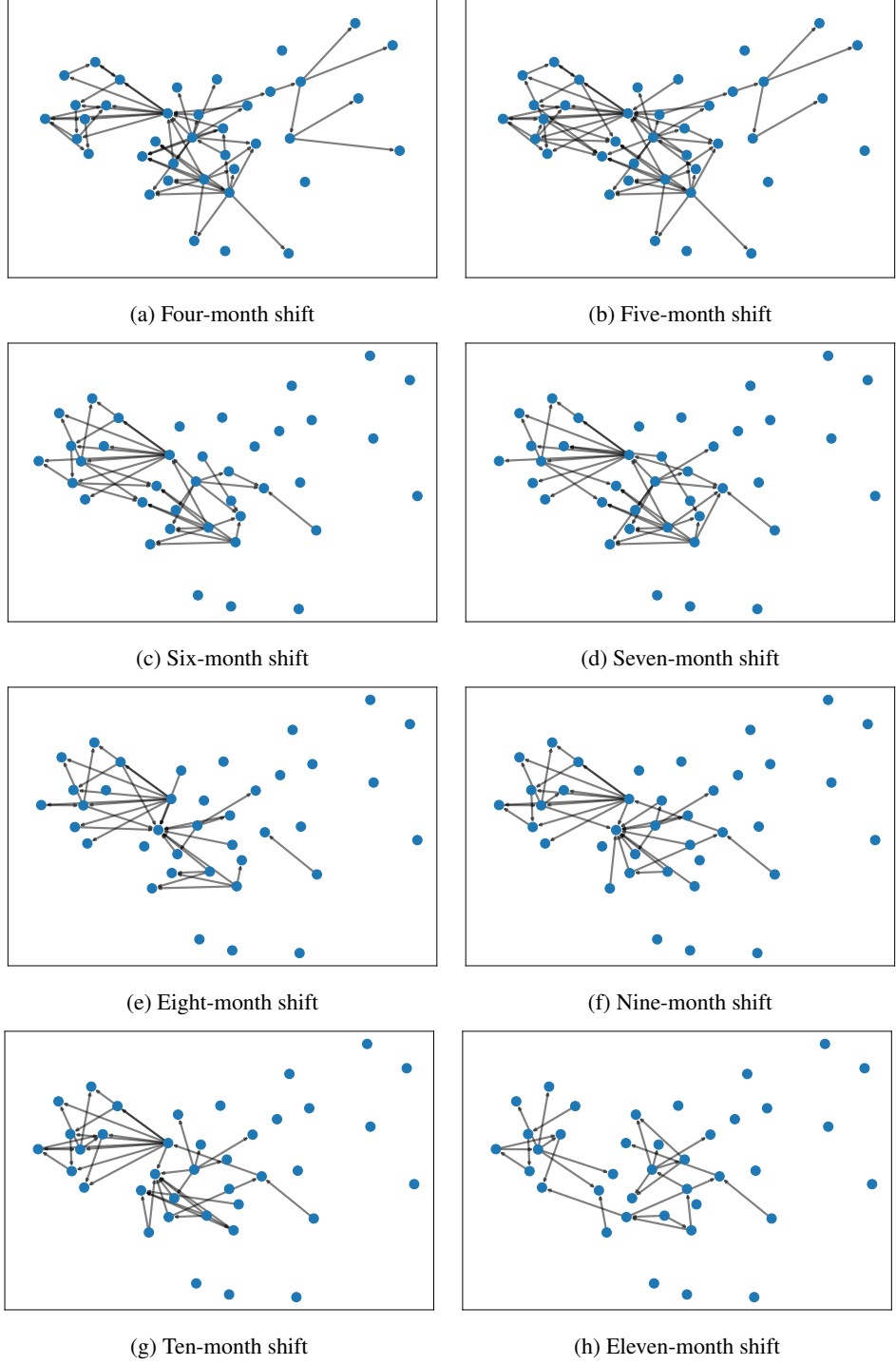

(a) Four-month shift

(b) Five-month shift

(c) Six-month shift

(d) Seven-month shift

(e) Eight-month shift

(f) Nine-month shift

(g) Ten-month shift

(h) Eleven-month shift

Figure 9: Evolution of the learned DAG on PMU-B over time.

