# OpenReview forum: "Graph-Augmented Normalizing Flows for Anomaly Detection of Multiple Time Series"
_ICLR.cc/2022/Conference — ICLR 2022 Spotlight_

### Official Review · Reviewer_ofqA · 2021-11-02

**Correctness:** 1
**Technical Novelty And Significance:** 2
**Empirical Novelty And Significance:** 1
**Recommendation:** 6
**Confidence:** 4

**Main Review:**

## Strength

The problem of anomaly detection from multivariate time series is relevant. In particular, anomaly detection in a fully unsupervised manner is technically challenging and an important research topic. Many existing methods (implicitly) assume availability of partial supervised information, such as normal profile, even if they are said to be unsupervised, while we cannot often assume such a situation in practice.

## Weakness

This paper is overall not well-written, and there are several concerns regarding presentation, quality and evaluation.

- First, I cannot understand the problem setting. The authors say that a give dataset is $(\mathbf{X}^1, ..., \mathbf{X}^n)$, and each $\mathbf{X}^i \in \mathbb{R}^{T \times D}$. In the typical setting of multivariate time series, there are $D$ each time series, and the length (the number of time stamps) of each time series is $T$. Then a given dataset can be treated as a single matrix $X \in \mathbb{R}^{T \times D}$. Therefore it is not clear why there are $n$ such matrices in the problem setting introduced in this paper? What is each $i$ here? Moreover, it is said that there are $|\mathcal{D}|$ such $(\mathbf{X}^1, ..., \mathbf{X}^n)$ as a dataset. Since repeated sampling is fundamentally impossible in time series analysis, it is also not clear what does such a dataset mean. In addition, the authors assume a DAG structure over such indices from 1 to $n$ ($n$ variables?), while its interpretation is also not clear. Please note that it makes sense if a DAG structure exists over $D$ time series or $T$ time stamps, while the setting in this paper is different.

- I am also confused with the anomaly measure. It is defined for each $\mathbf{X}^i$ as explained in Sec.5.2. However, the goal of anomaly detection for time series is usually to find anomalous regions of time stamps, and it seems that this task is actually considered in experiments (in Fig. 2b). Since the proposed measure cannot directly achieve the task, some clarification is required.

- There are also concerns regarding the experimental protocol. The authors say that a dataset is divided into training/validation/test sets. However, since the problem is unsupervised, such separation does not make sense, and one can just use the entire dataset excluding the ground truth labels for training and evaluate the prediction performance using the ground truth.

- Parameter sensitivity is not examined in experiments. Since parameter tuning is fundamentally difficult in unsupervised learning (as one cannot use CV), at least the sensitivity should be examined, otherwise it is not clear how the obtained results are useful and robust w.r.t. parameter changes.

- Figure 4 has almost no information in its current state. Illustration should be improved.


**Summary Of The Paper:**

This paper proposes to combine learning of the DAG structure between variables and learning of distribution using normalizing flows for unsupervised anomaly detection from multivariate time series. Once the distribution is learned, one can detect anomalies as low density regions. The effectiveness of the proposed method is empirically evaluated on real-world datasets.

**Summary Of The Review:**

Although this paper studies a relevant problem, the quality, presentation, and evaluation of this paper are not convincing.

---

> ### Author Response · Authors · 2021-11-22
> **Response to the Reviewer ofqA**
>
> Thank you for the comments. You seem to have misunderstood the setting we set forth. Let us clarify in what follows. We will be happy to further discuss should you have new questions.
>
> **Response to the problem setting**: In this paper, we focus on **multiple** multivariate time series: a data instance consists of $n$ multivariate times series (called constituent series), each of which is D dimensional (i.e., having D attributes) and has T time steps. The term “multiple time series” we used is standard in statistics and time series analysis. The task is to label such a data instance normal/abnormal. Many data instances are used to train a model. We have $|\mathcal{D}|$ such data instances for training.
>
> For example, a PMU dataset contains $n$ sensors located in different geographic locations, where each sensor recording is D dimensional (e.g., voltage, current, frequency, etc). The sensor recordings span a long period, which we cut into short windows, each of which contains T time steps. Such a window forms one data instance. The processing results in many data instances. We pick the first $|\mathcal{D}|$ for training, the next a few for validation, and the remaining for testing.
>
> This processing and setup are standard. The contribution we propose is to learn a graph among the $n$ constituent series.
>
> **Response to the anomaly measure**.
>  For each data instance, we compute its density based on a fitted parameterized distribution and this density is the anomaly measure. This density is for all n sensors. Because of the DAG factorization, we simultaneously obtain a conditional density for each sensor, and hence each conditional density serves as the anomaly measure for that sensor. For example, a grid event (anomaly) may be local, which means that besides the observation that the overall density is small, some sensors affected by the event also have small conditional density.
>
> **Response to the experiment protocol**: Our experiments follow standard machine learning practice: as aforementioned, we cut the time span into short windows and chronologically split the windows into training, validation, and test sets. The training set is used to learn the model, the validation set is used to find good hyperparameters, and the test set is used to report performance. The validation does not need supervised labels to perform, but it is performed in order to obtain optimal hyperparameters (such as when to stop optimization) and to avoid overfitting.
>
> **Response to the parameter tuning and sensitivity**: As aforementioned, we use the validation set to obtain good hyperparameters. They are bound to cause variations in the model performance, which is the reason why we need tuning. Tuning is also a means to mitigate overfitting: if the training loss keeps dropping but the validation loss bounces back, it is time to stop training.
>
> **Response on Figure 4**: Note that Figure 4 was renumbered to Figure 5 after we updated the paper. This figure shows a seasonal pattern of the appearance and disappearance of edges, indicating distribution drift.  The edges can loosely be grouped in three clusters: those persisting the entire year, those appearing in the first half of the year, and those existing more briefly (e.g., within a season). We believe the visualization is informative. It would be helpful if you have suggestions to further improve it.

---

> > ### Comment · Reviewer_ofqA · 2021-11-24
> > **Thank you**
> >
> > Thank you for your clarification. Now I understand the problem setting and the contribution of this paper. Hence I have raised my score to 6.
> >
> > About the problem setting, what confused me is the lack of explanation about the window based approach. I thought that T is the length (the number of time steps) of not a short window but the entire time series. Although this is briefly described in each dataset description, it would be better to also explicitly state in the problem statement.
> >
> >
> > I still do not fully understand the parameter turning. Of course I know the role of validation set, but I think it is a bit tricky as the current problem is unsupervised. In the turing of hyperparameters, how to evaluate the goodness of each hyperparameter setting on the validation set? Could you elaborate a bit more about this point?

---

> > > ### Author Response · Authors · 2021-11-28
> > > **Further Response to the Reviewer ofqA**
> > >
> > > Thank you for confirming the setting and contributions of our paper. Here are more details about hyperparameter tuning.
> > > - The proposed GANF aims to estimate the density of each data instance and identify anomalies with low densities. Therefore, an unsupervised density estimation metric on the validation set can be used to select the hyperparameters. Similar to the evaluation of normalizing flows in prior work, we use average log-density on the validation set to select the hyperparameters. The core idea of this widely-used unsupervised metric is that the learned model should assign high densities to normal instances that unseen in the training set.
> > > - Thus, we search the hyperparameters that yield the highest average log-density on the validation set. In addition, we stop training when the log density on the validation set begins to decline for several epochs to avoid overfitting. The parameters at the best validation epoch are stored as the final parameters.

---

> > > > ### Comment · Reviewer_ofqA · 2021-11-28
> > > > **Thank you**
> > > >
> > > > Thank you for your explanation. Then how do you know "normal instances" that are unseen in the training set? More precisely, anomalies also may exist in the validation set in the unsupervised setting, and one cannot know which data point is normal and which is anomaly. So if you select parameters with the highest average log-density on the validation set, it may also give high density to anomalies that exist in the validation set, which may be suboptimal. Is there any additional assumption such as anomalies do not exist in training and/or validation sets? Or can such an effect be negligible as the number of anomalies is usually much smaller than that of normal instances?

---

> > > > > ### Author Response · Authors · 2021-11-29
> > > > > **Further Response to the Reviewer ofqA**
> > > > >
> > > > > Thanks for the question. Here are a few points we’d like to clarify.
> > > > > 1.	We construct the validation set by selecting all data instances belonging to a certain period, non-intersecting with that of the training set. Thus, the validation loss (average negative log-density) is equivalent to the KL divergence between the true distribution and the flow recovered distribution, evaluated by using validation instances. This metric reflects the generalization of the model outside the training set. Thus, it measures how well the flow model is trained and it is a natural choice for tuning hyperparameters and guarding against overfitting.
> > > > > 2.	One cannot rule out anomalies in the validation set (as well as in the training set), but generally they are few. Thus, the effects of anomalies on the average log-density would be negligible. That is why we say “the core idea ... is [intuitively] that the learned model should assign high densities to normal instances”. Other metrics for hyperparameter tuning are possible. For example, one can also use the median of the log-density on the validation set, which reduces the effects of anomalies.

---

### Official Review · Reviewer_J4SV · 2021-11-02

**Correctness:** 3
**Technical Novelty And Significance:** 3
**Empirical Novelty And Significance:** 3
**Recommendation:** 8
**Confidence:** 4

**Main Review:**

Strengths:
- Good solution to a problem that has applications in many diverse areas
- Authors model correlations between time series, which is often ignored in multivariate time series work
- Authors effectively integrate mature components from a variety of different subfields in statistics, representation learning, and neural networks
- Despite all the different moving parts, the model is clearly explained
- Authors evaluate their methods in relevant datasets

Weaknesses:
- It is not clear how to quantitatively select the threshold for flagging something as an anomaly
- The chosen evaluation metric is difficult to interpret
- No discussion on scalability

Extended comments:
I found the presented model to be quite interested, and I believe that it addresses some of the weaknesses of existing anomaly detection methods in novel ways. Particularly, I believe that combining a Bayesian network with and RNN to jointly learn dependencies across time and constituencies adds great value to multivariate time series modeling. The fact that the model is able to learn the structure of the Bayesian network as a continuous optimization problem is quite advantageous as well.

I was also quite pleased with the presentation of the paper. Despite the fact that the authors are combining components from various ML and statistics subfields, the description is easy to follow.

Where I think the paper could improve is in the Experiments section. I understand that there is noise in ground truth labeling in anomaly detection datasets; however, the proposed evaluation metric to address this source of noise is difficult to interpret and frankly seems to be an afterthought. I would prefer to see a "hard" label; the authors could declare an anomaly as "successfully detected" if it falls within some time window of the prediction, where the size of that time window is application-specific.

Another weakness of the paper is on selecting the threshold to flag something as an anomaly. This is not a weakness specific to the paper, and potential solutions have been studied before. I think it would strengthen the paper if the authors propose heuristics to numerically choose a threshold, without having to pick and choose from a histogram (which is very subjective). Since there is a Gaussian distribution at the end of the model, would it be possible to say that an observation is anomalous if the probability of seeing such extreme values is less than 0.05 (or some other threshold)?

Finally, I would say that the ablation study is inconclusive. There is a marginal increase in performance; however, couldn't this be explained merely by the fact that GANF has more parameters (and hence more expressibility) than the models where you take out one of the components?

**Summary Of The Paper:**

The authors propose a model for anomaly detection on multivariate, high-dimensional time series data, where there are statistical dependencies across the different time series (referred to as "constituents" in the paper). The authors model statistical dependence using a Bayesian network and temporal dependence using an RNN. One key feature of the proposed model is that it simultaneously learns the structure of the Bayesian network AND a representation of the temporal state. The authors combine these two representations, which then gets used as input to a normalized flow technique, which makes computation of the likelihood of the data mathematically and computationally feasible.

The authors show that the proposed model has improved performance for the anomaly detection task compared to recently-proposed neural-network anomaly detection methods.

**Summary Of The Review:**

Given my comments above, I think that there are enough strong points in the paper to put it above the acceptance threshold. I am open to being more optimistic about it after the author response.

---

> ### Author Response · Authors · 2021-11-22
> **Response to the Reviewer J4SV. Part 2.**
>
> > Q3 “There is a marginal increase in performance; however, couldn't this be explained merely by the fact that GANF has more parameters?”
>
> Thank for the good question. Let us discuss each variant in more detail:
> * GANF\G: This variant eliminates the GNN component. The number of parameters therein is indeed fewer than GANF. However, GANF outperforms GANF\G by 4% and 7% on PMU-B and PMU-C, respectively. Such an improvement is noticeable.
> * GANF\D: This variant does not simplify the joint distribution as does DAG; all constituent series are connected before running the normalizing flow. Thus, the number of parameters therein is, to the contrary, way more than GANF. Even so, this variant is outperformed by GANF.
> * GANF\T: This variant uses an adjacency matrix pretrained by NOTEARS. Hence, the number of parameters therein is the same as GANF. The higher performance of GANF suggests that joint training of the adjacency matrix and other model parameters helps improve the performance.
> * GANF_RNVP: This variant only replaces the normalizing flow backbone from MAF to RealNVP. The choice of the flow backbone is not the focus of this paper; any flow can be used. The best flow appears to be case-dependent. As Table 2 shows, the two flows achieve comparable results, while being significantly better than other variants outside the flow choice. This observation in fact demonstrates the flexibility of our GANF framework.
>
> >Q4. “No discussion of scalability”.
>
> We have updated the paper with time complexity analysis in Appendix C. Note that Bayesian network structure learning is traditionally considered intractable, but a continuous (but equivalent) formulation makes it possible to apply gradient-based optimization techniques to efficiently obtain a good solution.

---

> > ### Comment · Reviewer_J4SV · 2021-11-29
> > **Updated score**
> >
> > I thank the authors for addressing my comments. I went through the revised draft and the responses to other reviewers. I have no further questions. The new results increase my confidence in the proposed method. I have updated my score accordingly.

---

> ### Author Response · Authors · 2021-11-22
> **Response to the Reviewer J4SV. Part 1.**
>
> Thank you for the constructive and insightful comments. We answer your questions in the following and we have updated the paper accordingly.
>
> > Q1 “The proposed evaluation metric to address this source of noise is difficult to interpret and frankly seems to be an afterthought. I would prefer to see a "hard" label; the authors could declare an anomaly as "successfully detected" if it falls within some time window of the prediction”
>
> Properly evaluating models against noisy ground truths is a rather tricky subject. We would like to elaborate on a few points here.
>
> First, we have updated the paper with an additional dataset (SWat) which provides accurate, hard labels, in part to mitigate the controversy on noisy labels. The AUC scores were added to Table 1 and the ROC curves were plotted in Figure 2. The curves were computed by using the standard definition of TP/FP rates. We observe similar results to the PMU datasets: our GANF model noticeably outperforms all baselines.
>
> Second, for the PMU datasets, we were notified of the noisy (and incomplete) labeling by data providers in the first place. Hence, adapting the standard definition of TP/FP rates to these datasets is practically needed. In a sense, we are using the windowing technique as you suggested: the sigma bandwidth in the Gaussian-like probability definition of a ground truth specifies the window (except that not every time step inside the window is treated as a ground truth anomaly with 100% probability). We believe this definition is fair for both positive and negative samples: a low-density sample will not be treated as a false negative merely because it is not exactly aligned with the labeled time step; and a high-density sample will not be treated as a false positive merely because it falls inside the window but otherwise does not coincide with an actual anomaly.
>
> > Q2. “Since there is a Gaussian distribution at the end of the model, would it be possible to say that an observation is anomalous if the probability of seeing such extreme values is less than 0.05 (or some other threshold)?”
>
> Thanks for the suggestion. However, we clarify two points.
>
> First, a low density in the latent Gaussian does not necessarily mean a low density in the original data space, and vice versa. One intuition comes from the change-of-variable formula $\log⁡{p(x)}=\log{⁡q(z)}+\log⁡ |\nabla⁡_x f(x) |$, which indicates that a small Gaussian density $q(z)$ does not indicate a lower data density $p(x)$, because the Jacobian term may vary. Another intuition comes from visualizing a mixture of two Gaussians. If the two Gaussians are reasonably but not overly separated, the low-density region between them may be mapped to the middle of the latent Gaussian (which suggests high density), depending on what flow mapping one obtains.
>
> Second, applications may have different priorities on success; hence, setting a threshold encourages gaming against the priority, causing unfair comparisons. For example, in cases where a higher true positive rate is more desired, one may let a model report as few anomalies as possible, solely for the sake of beating competitors but otherwise fail to identify more anomalies as the model should. This is exactly the reason to use a ROC curve to investigate all possible thresholds. We have added such curves in Figure 2 of the updated paper.

---

### Official Review · Reviewer_tq1S · 2021-11-03

**Correctness:** 4
**Technical Novelty And Significance:** 3
**Empirical Novelty And Significance:** 3
**Recommendation:** 6
**Confidence:** 4

**Main Review:**

Strengths of the paper:
(1) The paper is clearly written and describes a novel technique. The combination of the DAG and the normalizing flow is a very interesting approach, and the application to the multiple time-series anomaly detection problem is a novel contribution.

(2) For the analyzed dataset, the proposed method appears to significantly outperform the baselines.

(3) The authors provide an ablation study to demonstrate that all of the components of the algorithm contribute to the improved performance.

Weaknesses of the paper:
(1) The quantitative results are only derived for one dataset (which is unfortunately proprietary). The labels for this dataset are noisy, making the significance of the results more difficult to assess. The experiments on the traffic dataset seem incomplete and it is difficult to conclude much from them.

(2) There are no measures of variability or confidence intervals for the provided results. There are no tests to determine if the performance differences are statistically significant. The authors calculate AUC-ROCs, but do not display the ROCs, meaning that the comparison to baselines boils down to a single number.

(3) The absence of synthetic data or well-understood data means that it is difficult to determine whether the derived DAGs are meaningful.

MAIN REVIEW

The paper addresses a very interesting and challenging problem and provides an elegant and intriguing solution. The combination of the RNN, the DAG to capture the dependency structure, and the normalizing flow appears to be a promising way to represent multiple time-series. The experimental results suggest that the joint learning, with enforcement of an acyclic graph, can lead to better performance.

Methodologically, I think the paper presents a novel, technically sound, and well-motivated approach. There have been some recent approaches in the multivariate time-series forecasting literature to combine latent representations and normalizing flow, but I am unaware of any work that incorporates a DAG in a similar way.

The experiments are the weaker portion of the paper.

(1) I think the paper would have benefited significantly from the inclusion of an analysis of synthetic data. With control over the ground truth, the performance of the algorithm could be explored and better understood. In particular, one could assess how well the DAGs are being estimated.

(2) In the absence of synthetic dataset, the experiments would be considerably more compelling if more datasets were analyzed (preferably with at least one public dataset).

(3) It is highly desirable that some form of confidence interval (or alternative measure of variability) is provided for the obtained performance. Statistical significance tests should also be conducted.

(4) Compressing the performance metrics to a single number is undesirable. I would have been very interested to see a comparison of the actual estimated ROCs. Often the AUC can give a misleading picture because one is more interested in relative performance in the low false-alarm regime. Figures such as 2(b) are very useful, but the paper does not provide the equivalent figures for any baseline algorithms, making comparison difficult.

(5) More details about hyperparameter tuning process would be welcome. The statement “The hyperparameters are all tuned based on the log-density on the validation set” is not sufficient to reproduce the results unless it is clear which hyperparameters were tuned, how they were tuned and over what ranges. For example, it is not clear if all architectural choices are included (hidden dimensions, number of flow blocks)

(6) The experiments for the traffic data seem incomplete, and it is difficult to draw much of a conclusion from them.

Questions:

(1) Is it possible to provide some form of measure of the variability in the results, e.g. bootstrapped confidence intervals?

(2) Some of the forecasting architectures come closer to having an A matrix that is applied to previous states, i.e. AH_{t-1} W_2. With this form there is no need to enforce a DAG structure, but predictive relationships are learned. There may also be sparsity encouragement during the training process. I thought this form might be one of the ablation studies to demonstrate the importance of the modeling of contemporaneous dependencies, but my understanding was that the two ablation cases investigating alternative dependencies were different from this approach. Could you comment – do you have insight into the importance of learning a contemporaneous as opposed to a predictive A? Please clarify if my understanding of one of the ablation settings is wrong.


**Summary Of The Paper:**

The paper addresses the task of detecting anomalies in multiple time-series, for the setting where multiple instances of a fixed time-window are available for learning. The authors propose a method that involves using a directed acyclic graph (DAG) to model the (contemporaneous) dependencies between the variables in the time-series. An RNN is applied to form a representative state for each time-series at each point in time. A mapping is applied using the DAG to process the states from the previous time-step and the ancestor states from the current time-step to construct a “dependency representation” vector for each time series. A normalizing flow is then applied to map to a base distribution so that the log-density of the instance (the observed multiple time-series) can be evaluated. The conditional densities for each time series can also be evaluated. The paper includes an evaluation of the performance of the approach on a proprietary dataset, with a comparison to state-of-the-art baselines. Some qualitative results are also included for the task of density estimation for a public traffic dataset.

**Summary Of The Review:**

** After the authors' response and the improvements to the paper, I have changed the score to a "6"; marginally above the acceptance threshold.

-----------------------------------------------------
I have recommended “marginally below the acceptance threshold”.


I like the methodology that is presented in the paper, but I think there is a need for more extensive experimentation. Synthetic data would provided a much more controlled testing environment. An examination over more datasets would provide more compelling evidence that the proposed method is advantageous, and perhaps highlight the types of problems where the approach provides a major benefit, as opposed to those where it offers a less significant improvement (or is perhaps even detrimental due to its flexibility). More details concerning the experimental methodology are required. Results that provide a more detailed comparison between the proposed methods and baselines would be helpful (some figures rather than just a single number). Measures of variability in the results and statistical significance tests would be beneficial in understanding how dramatic an improvement has been made.

---

> ### Author Response · Authors · 2021-11-22
> **Response to the Reviewer tq1S**
>
> Thank you for agreeing with the novelty and contributions of our work. We address your concerns in the following and we have updated the paper accordingly.
>
>
> > Q1. “Is it possible to provide some form of measure of the variability in the results?”
>
> Yes, we have added standard deviations to Table 1, through repeatedly training models multiple times. All models, including the baselines, suffer randomness in one way or another; for example, our model is trained by using stochastic gradient descent and hence model performance (slightly) varies. However, the variability is substantially smaller than the gap between our model’s performance and those of the baselines. Additionally, we have performed significance tests and confirmed that GANF’s results are significantly better than those of the baselines statistically (p<0.001).
>
> > Q2. “In the absence of synthetic dataset, the experiments would be considerably more compelling if more datasets were analyzed (preferably with at least one public dataset).”
>
> We have added experiments on a public dataset SWaT, which regards secure water treatment, where ground-truth anomalies were generated through system attack. We included the performance numbers in Table 1. The results are in line with those on the power grid datasets; they suggest that our method outperforms all baselines.
>
> | EncDecAD | DeepSVDD | ALORCC | DROCC | DeepSAD | Our GANF |
> |  ----------- | ----------- | ----------- | ----------- | ----------- | ----------- |
> | 76.5±0.7 | 68.8±2.0 | 75.4±2.3 | 73.3±1.6 | 75.4±1.2 | **79.6**±0.9 |
>
>
> > Q3. “I would have been very interested to see a comparison of the actual estimated ROCs”
>
> Thanks for the suggestion. We have updated the paper by adding ROC curves (see Figure 2). The GANF curves generally dominate other curves, suggesting that GANF outperforms the baselines not only in a single AUC score but also for most of the thresholds.
>
> > Q4. “More details about hyperparameter tuning process would be welcome”
>
> In experiments, we tuned the following hyperparameters: learning rate, hidden dimension, and the number of normalizing blocks. We conducted grid search to find the parameters that yield the highest log-density on the validation set. We updated the paper by specifying the tuning range (see Appendix B). In addition, we will release the code for reproducibility.
>
> > Q5. “Some of the forecasting architectures come closer to having an A matrix that is applied to previous states, i.e. AH_{t-1} W_2. With this form there is no need to enforce a DAG structure, but predictive relationships are learned. There may also be sparsity encouragement during the training process. I thought this form might be one of the ablation studies to demonstrate the importance of the modeling of contemporaneous dependencies”
>
> The matrix $\mathbf{A}$ corresponding to a DAG is the vehicle for us to factorize the joint distribution of all graph nodes, so that we can compute probabilities and can have an unsupervised training. For ease of discussion, let us write $\mathbf{D} = \mathbf{A}\mathbf{H_{t-1}}\mathbf{W_2} $, although there are additional terms that do not matter. With $\mathbf{A}$ corresponding to a DAG, the i-th row of $\mathbf{D}$ depends on only those rows of $\mathbf{H_{t-1}}$ associated to the parents of i, making the density calculation formula (9) valid.
> Without the DAG structure, the matrix A models contemporaneous dependencies like in VAR models. In this case, a formula like $\mathbf{D} = \mathbf{A}\mathbf{H_{t-1}}\mathbf{W_2} $ can be treated as a feature extractor to compute higher-level representations of the data $\mathbf{X_{t-1}}$. This is nothing but a graph convolutional network (GCN), and we can do graph convolutions for all time steps t. The remaining question is what to do with the representations extracted by GCN. If we follow our paper’s idea, some form of normalizing flow will be needed, which could be the already ablated variant GANF\D in Table 2 (through concatenating the constituent series along the attribute dimension). However, model GANF\D by itself already models the interdependencies among graph nodes because of running MAF on the concatenated data. Hence, modeling the same concept (contemporaneous dependencies) twice appears redundant.

---

> > ### Comment · Reviewer_tq1S · 2021-11-25
> > **Assessment of revised paper and author's response**
> >
> > I thank the authors for their response. I read it carefully and also consulted the other reviews and responses.
> >
> > Overall, I think the paper has improved considerably, and I have raised my score accordingly.
> > (1) The extra public dataset provides useful support for the method and will allow other researchers to conduct comparisons.
> > (2) Thank you for adding the standard deviations for the results. Since you conducted statistical significance tests, it would be good to include a brief description of the process for testing statistical significance (e.g., are you making normality assumptions and are they justified?) and to include a record of the outcomes in the paper.
> > (3) The ROCs provide valuable insights into the behaviour of the algorithm, and make it much easier to compare in a meaningful way with the baselines.
> > (4) Thank you for providing more information about your hyperparameter tuning procedure. The details about the baselines are still too vague: “All hyperparameters of the baselines are tuned based on the validation set to make fair comparisons.”
> > (5) The experiments with the traffic set still seem incomplete, to the extent that advertising them in the main part of the paper as "We
> > conduct experiments on ... one traffic dataset" appears misleading to me, giving the impression that the experiments are of the same quality or nature as those conducted for the other datasets.

---

> > > ### Author Response · Authors · 2021-11-28
> > > **Further Response to the Reviewer tq1S**
> > >
> > > Thank you for agreeing with the improvements that we made. Here are more details that address your further comments:
> > >
> > > > Q1“Since you conducted statistical significance tests, it would be good to include a brief description of the process for testing statistical significance (e.g., are you making normality assumptions and are they justified?) and to include a record of the outcomes in the paper.”
> > >
> > > Thanks for the question and suggestion. Here is the procedure we take for the significance tests:
> > > 1. We first conduct the Shapiro-Wilk test for normality.
> > > 2. If the null hypothesis is rejected, we apply the Mann–Whitney U test to check whether our GANF is significantly better. Otherwise, we apply Student’s t-test.
> > >
> > > Following the above procedure, we obtained that all p-values of the hypothesis tests are less than 0.001. This shows that our GANF is significantly better than the baselines.  We will add the description and the results of the hypothesis tests in the paper.
> > >
> > > > Q2. “The details about the baselines are still too vague: ‘All hyperparameters of the baselines are tuned based on the validation set to make fair comparisons’.”
> > >
> > > Here are more details about the hyperparameter settings of the baselines:
> > >
> > > For EncDecAD and DeepSVDD, we tune the learning rate. For ALOCC, we tune the learning rate and the parameter that balances the reconstruction loss and the adversarial loss. For DROCC, apart from the learning rate, we tune the weight of the loss on adversarial samples. For DeepSAD, we vary the learning rate and the contribution of the semi-supervised loss. The architectures of these models are either RNN or one-dimensional CNN. The hidden dimensions of all baseline models are set the same as our GANF for fair comparisons.

---

### Official Review · Reviewer_mBgF · 2021-11-04

**Correctness:** 3
**Technical Novelty And Significance:** 4
**Empirical Novelty And Significance:** 3
**Recommendation:** 8
**Confidence:** 3

**Main Review:**

I liked this paper and read it with great interest. The strength of this paper rests in a) an excellent problem setup, and b) well chosen methods that address real challenges in the field and provide useful insights into problems. The paper does not make unrealistic assumptions. The evaluation datasets are well chosen and the baselines do not seem like "strawman" comparisons.

I am particularly impressed with Figure 4. Finding the temporal shifts for anomalies, in particular anomalies that develop slowly is a valuable contribution, and being able to quantify that with with a Bayesian graph is a novel contribution that deserves to be called out. Overall, I would be interested to hear more about the explainable side of this work - Bayesian networks are often useful for counterfactual reasoning which makes me optimistic that these methods are not only useful for detecting anomalies, but also potentially explainable.

There are some places where I think the evaluation could be improved. ROC-AUC is useful real-valued metric, but in practice, there are always questions about the choice of threshold and inclusion of of some ROC curves, along with uncertianty bounds on those curves would make this work even stronger.

This paper also suffers from complexity and I somewhat question its reproducability. I don't doubt the numerical results, but with a complex system such as this one, there would need to be a significant investment of effort if anyone else wanted to use these methods and I don't think sufficient details are provided in . Providing a code repository would address this.

If I have a major complaint about this paper it is that their main hypothesis: "Anomalies lie on low density regions of the data distribution" remains untested.

**Summary Of The Paper:**

This paper focuses on unsupervised anomaly detection in multivariate time series. This is a important problem because anomalies are rare and labeling anomalies for supervised learning is labor intensive. To make progress in this challenging domain, they  use learned graph structure and normalizing flows, which seem well suited for the task. They evaluate their methods on multiple datasets, and compare against multiple modern deep networks.

**Summary Of The Review:**

This is a strong paper and I recommend it for acceptance. It addresses a real problem in a manner that is both novel and pragmatic. I hope the authors will not take my constructive criticism too harshly, the paper is strong in its current form and my suggested improvements are only suggestions for ways I think it could be made even stronger. I would love to see this work see application in other areas and my comments reflect ways to speed that process.

---

> ### Author Response · Authors · 2021-11-22
> **Response to the Reviewer mBgF**
>
>
> Thank you for the detailed and insightful comments. We have updated the paper based on your suggestions.
>
> 1. **Response to explainability**: Indeed, an advantage of using the Bayesian network formality for graph structure learning is that the relationship between graph nodes can be understood from conditional (in)dependence. The interpretation can be more easily carried out in applications where node metadata is available. For example, for the power grid, were spatial information of the PMUs known, the identified DAGs would help map the seasonal patterns of the edges to geography and shed light on the propagation of failures, which in turn helps forge a more resilient grid.
>
> 2. **Response to improving evaluation**: As you suggested, we have updated the paper with ROC curves (see Figure 2). Additionally, for the AUC sores, we added standard deviations through repeating model training multiple times (see Table 1). All models suffer randomness in one way or another; for example, our model is trained by using stochastic gradient descent; hence, model performance (slightly) varies. However, the variability is substantially smaller than the gap between our model’s performance and those of the baselines. Furthermore, we have included an additional dataset (SWaT) and it similarly shows the superiority of our model.
>
> 3. **Response to reproducibility**: We have included hyperparameter tuning details in the updated paper (see Appendix B). We will release the code when the paper is accepted.
>
> 4. **Response to the validation of the hypothesis**: We hope that our good detection results in a sense validate the otherwise intuitive hypothesis that outliers have low density. Intriguingly, for the power grid, data providers believe they have not fully annotated all grid events. Hence, the additional low-density points we identified raise interest to power system experts for further analysis.

---

### Author Response · Authors · 2021-11-22
**General response to all reviewers and updates of the paper**

We thank all reviewers for your efforts to read the paper and feel glad about your generally favorable assessment of the methodology development (we respond separately to a reviewer who appears to have misunderstood our setting). We have updated the paper (with text highlighted in red), focusing on enriching the experiment results. Here is a summary of the updates:
-	Added a dataset with ground truth labels for evaluation.
-	Added ROC curves.
-	Added variability of the AUC scores.
-	Added hyperparameter tuning details (see appendix).
-	Added time complexity analysis (see appendix).
-	Moved the exploratory analysis of the traffic dataset to the appendix.

---

### Public Comment · ~Kashif_Rasul1 · 2022-04-27
**Figure 1 typo**

In figure 1 I believe there is a superscript i missing? should be $\mathbf{x}^i_{1:t-1}$

Another question I had was, why can't this model be used for the prediction task? Thanks for any insights!

---

> ### Public Comment · ~Jie_Chen1 · 2022-04-29
> **Thank for you catching the typo!**
>
> Will update the figure when OpenReview is re-open.

---

> > ### Public Comment · ~Jie_Chen1 · 2022-05-26
> > **Figure updated.**
> >
> > Thanks!

---

### Decision · Program_Chairs · 2022-01-20

**Decision:**

Accept (Spotlight)

**Comment:**

The paper tackles the problem of detecting anomalies in multiple time-series. All the reviewers agreed that the methodology is novel, sound and very interesting. Initially, there were some concerns regarding the experimental evaluation, however, the rebuttal and subsequent discussion cleared up these concerns to some extent and all reviewers are eventually supporting or strongly supporting acceptance.